# Vardenafil Activity in Lung Fibrosis and In Vitro Synergy with Nintedanib

**DOI:** 10.3390/cells10123502

**Published:** 2021-12-11

**Authors:** Michael H. Bourne, Theodore J. Kottom, Deanne M. Hebrink, Malay Choudhury, Edward B. Leof, Andrew H. Limper

**Affiliations:** Thoracic Diseases Research Unit, Departments of Medicine and Biochemistry, 8-24 Stabile, Mayo Clinic, Rochester, MN 55905, USA; bourne.michael@mayo.edu (M.H.B.J.); kottom.theodore@mayo.edu (T.J.K.); hebrink.deanne@mayo.edu (D.M.H.); choudhury.malay@mayo.edu (M.C.); leof.edward@mayo.edu (E.B.L.)

**Keywords:** vardenafil, PDE5, fibrosis, TGF-β1

## Abstract

Idiopathic pulmonary fibrosis (IPF) remains an intractably fatal disorder, despite the recent advent of anti-fibrotic medication. Successful treatment of IPF, like many chronic diseases, may benefit from the concurrent use of multiple agents that exhibit synergistic benefit. In this light, phosphodiesterase type 5 inhibitors (PDE5-Is), have been studied in IPF primarily for their established pulmonary vascular effects. However, recent data suggest certain PDE5-Is, particularly vardenafil, may also reduce transforming growth factor beta 1 (TGF-β1) activation and extracellular matrix (ECM) accumulation, making them a potential target for therapy for IPF. We evaluated fibroblast TGF-β1-driven extracellular matrix (ECM) generation and signaling as well as epithelial mesenchymal transformation (EMT) with pretreatment using the PDE5-I vardenafil. In addition, combinations of vardenafil and nintedanib were evaluated for synergistic suppression of EMC using a fibronectin enzyme-linked immunosorbent assay (ELISA). Finally, the effects of vardenafil on fibrosis were investigated in a bleomycin mouse model. Our findings demonstrate that vardenafil suppresses ECM generation alone and also exhibits significant synergistic suppression of ECM in combination with nintedanib in vitro. Interestingly, vardenafil was shown to improve fibrosis markers and increase survival in bleomycin-treated mice. Vardenafil may represent a potential treatment for IPF alone or in combination with nintedanib. However, additional studies will be required.

## 1. Introduction

Idiopathic pulmonary fibrosis (IPF) is a chronic, progressive, fibrosing interstitial lung disease (ILD) of unknown etiology with a poor prognosis, resulting in a median survival of 3.8 years in adults 65 years of age and older in the US [1,2]. The disease occurs worldwide, and its prevalence seems to be increasing, with an incidence between 3 and 9 cases per 100,000 person years in Europe and North America. Pathologically, the disease is characterized by the accumulations of extracellular matrix (ECM) components, including collagen and fibronectin associated with myofibroblast proliferation in areas of alveolar damage [3,4]. This loss of functional alveolar units leads to decreased lung compliance, impaired gas exchange, and ultimately respiratory compromise and death [5]. In addition, epithelial mesenchymal transformation (EMT) has been proposed as one mechanism to further promote the fibroproliferative process in this disease [6]. The pathophysiology of the disease remains complicated and multi-faceted, with many potential therapeutic targets [7].

The pathogenesis of idiopathic pulmonary fibrosis has not been fully elucidated, but it centrally involves epithelial injury, followed by fibroblast repair driven by TGF-b1 [3]. In addition to fibroblasts, other cells participate in the repair process during IPF, including epithelial cells, macrophages, and cells involved in epithelial mesenchymal transformation [3,4,6,7]. IPF is a disease that is strongly linked to the aging process, and cellular senescence also participates in the overall disease pathogenesis [3,4,6,7]. As an initial analysis, the current study focused on the effects of vardenafil on TGF-b1-driven fibroblast responses and the ability of vardenafil pretreatment to alter the measures of fibroproliferative repair.

Despite the recent approval of drugs such as nintedanib and pirfenidone, which have been shown to reduce the rate of lung function decline in IPF, patients with this intractable disorder continue to lose lung function and have poor long-term prognoses [8,9]. Treatment remains a burden to patients and the health care system in general, with high annual costs for the available anti-fibrotic medications pirfenidone and nintedanib, which also are also limited by frequent intolerable side effects and disease progression in most patients [10]. Accordingly, ongoing work is being performed to identify novel therapeutics [7,10,11].

Similar to multi-drug treatments for cancer, heart diseases, and HIV, the concept of combination therapies in IPF has recently been proposed as a therapeutic approach that may improve patient outcomes [12,13]. Previous studies have examined certain combinations such as prednisone, azathioprine, and *N*-acetylcysteine, but in this case, combination therapy was actually found to be harmful [14]. Since effective monotherapies including nintedanib and pirfenidone are now available, future trials should center on adding additional medications to these existing therapies. The combination of the two existing therapies, nintedanib and pirfenidone, has been investigated in the INJOURNEY trial, and combination therapy was found to have an acceptable safety and tolerability profile [15]. Further studies are needed to evaluate its efficacy.

Alternative agents, such as phosphodiesterase type 5 inhibitors (PDE5-Is), have been studied in IPF [16]. These agents have been used effectively in the treatment of pulmonary arterial hypertension and have had some reports of successful clinical responses in patients with scleroderma-associated pulmonary fibrosis, believed to be related to their effects on the lungs’ blood flow [17]. Others studies have documented improvements in the 6-min walk distance of patients with pulmonary hypertension associated with IPF [18,19]. In that light, a large multi-center trial of the PDE5-I sildenafil was performed in IPF patients [16]. Unfortunately, the IPF-STEP trial comparing sildenafil to a placebo in patients with well-characterized advanced IPF failed to demonstrate significant improvement of 20% or greater in the 6-min walk distance. Despite not reaching the primary outcome, small but significant differences were noted in arterial oxygenation, carbon monoxide diffusion capacity, degree of dyspnea, and quality of life favoring the sildenafil treatment [16]. A recent study of the combination of nintedanib plus sildenafil versus nintedanib alone in patients with right heart dysfunction (INSTAGE) did not demonstrate differences in the changes in the FVC and St. George’s Respiratory Questionnaire scores in the treatment groups [20].

Most of the potential therapeutic benefits of PDE5-Is have been attributed to their activity in the pulmonary vasculature, increasing blood flow to reduce exercise-exacerbated gas exchange abnormalities and improving dyspnea and exercise capacity. However, certain PDE5-I drugs such as vardenafil may actually exert direct effects on lung mesenchymal cells [21]. For instance, TGF-β1-induced generation of extracellular matrix proteins by lung fibroblasts and transformation of pulmonary epithelial cells to myofibroblasts are features of IPF and additional potential targets of PDE-Is. Studies have demonstrated PDE-I inhibition of TGF-β1-induced generation of ECM, as well as differentiation of lung fibroblasts to myofibroblasts in vitro [21].

In this light, we sought to determine whether the longer-acting PDE5-I vardenafil would inhibit TGF-β1-induced fibroproliferation in cultured fibroblasts by analyzing ECM mRNA and ECM protein upregulation in vitro, as well as epithelial mesenchymal transformation and TGF-β1 signaling. We then assessed whether combination doses of vardenafil and nintedanib would exert synergistic inhibition of TGF-β1-induced ECM production by cultured fibroblasts. Lastly, we tested the effects of vardenafil in the bleomycin mouse model of lung fibrosis. As a proof of concept, vardenafil was administered to both cell cultures and mice shortly after pretreatment with TGF-β1 (cells) or bleomycin (mice). As such, these studies demonstrate potential activity in fibrosis prevention. Herein, we demonstrate that vardenafil inhibits TGF-β1-driven ECM production in cultured lung fibroblasts both alone and synergistically with nintedanib, and it improves fibrosis and survival in a bleomycin-treated mouse model of the disease.

## 2. Materials and Methods

### 2.1. Animal and Cell Sources

All animal studies were justified and approved by the Mayo Institutional Animal Care and Use Committee. Eight-week-old female C57 black mice (~20 g; Jackson Laboratories, Bar Harbor, ME, USA) and several murine and human cell lines were used in this study. IMR-90 human and AKR-2B murine fibroblasts were used as previously reported [22]. The human IPF fibroblasts were from LONZA, Inc., Basel, Switzerland These fibroblasts were obtained by a primary culture from the lung explant of a patient undergoing lung transplantation for IPF and were used at passage numbers of four or less. The diagnosis of IPF was confirmed in each donor by a multidisciplinary group of clinicians, radiologists, and pathologists with specific expertise in IPF.

### 2.2. Chemical and Antibody Sources

PDE5-I vardenafil was obtained from Sequoia Research Products Ltd. (vardenafil citrate), Sigma Aldrich, or Selleckchem (vardenafil HCl). Nintedanib was obtained from Cayman Chemical. TGF-β1 was obtained from R&D Systems. The antibodies used in this study were as follows: fibronectin (Sigma Aldrich, F3648, St. Louis, MO, USA), CTGF (Santa Cruz Biotechnology, sc-14939, Dallas, TX, USA), αSMA (Sigma, A2547), p-SMAD3 (Cell Signaling 95205, Danvers, MA, USA), total SMAD3 (Abcam AB 28379), and GAPDH (Millipore MAB374). The secondary antibodies used for fibronectin ELISA included anti-rabbit IgG peroxidase (Sigma Aldrich, A0545) and anti-rabbit IgG-HRP (SantaCruz Biotechnology sc-2004). Additional antibodies for TGF-β1 signaling included the following: anti-mouse serpin E1/PAI-1 antibody (AF3828; R&D systems), anti-CTGF (sc-14939; SCBT), anti-αSMA (A2547; Sigma), anti-fibronectin (F3648, Sigma), anti-GAPDH (AB2302; Millipore, Burlington, MA, USA), anti-phospho-SMAD3 and anti-phospho-SMAD2 antibodies generated in our laboratory, anti-SMAD2 (ab63576; Abcam, Cambridge, UK), anti-SMAD3 (ab28379; Abcam), anti-phospho-AKT (Ser473; 9271, Cell Signaling), anti-phospho-AKT (T308; 4056, Cell Signaling), anti-phospho S6K (Thr389, 9234, Cell Signaling), anti-Akt (9272, Cell Signaling), and anti-S6K (9202, Cell Signaling).

### 2.3. Experimental Procedures

Details regarding the various experimental procedures investigating the anti-fibrotic effects of vardenafil are available in the online Appendix A. These methods included gene expression analysis by qPCR quantitation of COL1A1, fibronectin, and TSP-1 in IMR-90 lung fibroblasts and studies assessing the activity of vardenafil in suppressing anchorage-independent growth of fibroblasts. Other experiments, including investigating potential signaling mechanisms by which vardenafil may inhibit fibrosis via western blot as well as study methods for assessing potential synergistic drug activity between vardenafil and nintedanib in AKR-2B and human IPF fibroblasts with fibronectin ELISA, are described therein as well. Finally, in vivo studies using a bleomycin mouse model are also described. All cell culture conditions were analyzed against the control state, where the cells were cultured identically in the absence of TGF-β1 [22]. The IPF fibroblast lines were primary-cultured cells. While age-matched normal lung fibroblasts would serve as interesting controls, such age-matched normal cells are challenging to obtain. Accordingly, with IPF cells, we again internally compared the cell cultures in the presence and absence of TGF-β1 [22].

### 2.4. Data Analysis and Statistics

All data are presented as the mean ± SEM. Quantitative PCR data are shown as relative units after control background subtraction and normalization to maximal stimulation, defined as 100. Fibronectin ELISA data are also displayed as relative units after control background subtraction and normalization to maximal stimulation, defined as 100. In the bleomycin animal studies, based upon our prior work, we tested initial group sizes of 8–10 animals under each condition. This number of animals demonstrated sufficient power to detect fibrotic effects in the treated animals. With respect to the in vitro cellular work, initial pilot experiments were performed to identify the potential effects and determine appropriate dose ranges of the reagents and the cellular concentrations required. The final assays’ conditions were then tested for each principal finding from multiple experiments in different days, with each individual measurement being performed at least in triplicate. For all experimental analyses, the datasets were tested for normality using the Kolmogorov–Smirnov test. For normally distributed multi-group data, an initial analysis was first performed with ANOVA to determine the overall differences, with subsequent subgroup analysis performed by Neuman–Keuls multiple comparisons testing. For non-parametric data, initial multi-group testing was performed with the Kruskal–Wallis test, with subsequent use of Dunn’s multiple comparison testing for subgroup comparisons. The survival comparisons were analyzed using log rank Mantel–Cox testing. Statistical analysis was performed using Prism version 5.0b software (GraphPad, Inc.), and differences were considered to be statistically significant when *p* < 0.05.

## 3. Results

### 3.1. Vardenafil Inhibited TGF-β1-Mediated mRNA Levels of Key TGF-β1 Responsive Genes in Cultured Lung Fibroblasts

Since vardenafil has been previously indicated to inhibit TSP-1 and ECM generation, we sought to determine whether vardenafil would suppress matrix gene expression in cultured lung fibroblasts [23,24]. To address this, the levels of collagen type I, alpha 1 (COL1A1), fibronectin, and thrombospondin (TSP-1) mRNA were measured in IMR-90 fibroblasts using qPCR (Figure 1). At the higher doses tested, all three mRNA transcripts were significantly reduced compared with TGF-β1 pretreatment alone (Figure 1). Cell toxicity from the vardenafil was excluded by an XTT viability assay (Appendix A). Thus, these data strongly support the potential anti-fibrotic role of PDE5-I vardenafil in reducing extracellular matrix accumulation in lung fibroblasts.

### 3.2. Vardenafil Inhibited Morphological Transformation and Anchorage-Independent Growth (AIG) of Fibroblasts

We next assessed TGF-β1-driven morphology changes in the AKR-2B fibroblasts. AKR-2B cells were employed because they have been recognized to demonstrate TGF-β1-induced morphological changes and TGF-β1-stimulated, anchorage-independent growth, a phenotype of autonomous proliferation [25]. The AKR-2B fibroblasts treated with TGF-β1 morphologically transformed, with the appearance of elongated spindle-like projections. Pretreatment with vardenafil, however, inhibited these morphologic changes (Figure 2A). In addition, vardenafil also significantly suppressed the anchorage-independent growth of TGF-β1-stimulated AKR-2B fibroblasts (Figure 2B). Taken together, these data further support that vardenafil can suppress the fibroproliferative phenotype of fibroblasts in culture.

### 3.3. Vardenafil Substantially Reduced ECM-Related Protein Production by Lung Fibroblasts, but Had Little Effect on TGF-β1-Induced Increases in SMAD2 or 3 Phosphorylation

We further assessed the effects of vardenafil on the TGF-beta responsive proteins involved in matrix production and maintenance in AKR-2B fibroblasts. These target proteins included fibronectin, collagen IV (Col IV), connective tissue growth factor (CTGF), and plasminogen activator inhibitor-1 (PAI-1). These studies demonstrate that vardenafil suppressed the TGF-β-stimulated expression of profibrotic targets, including Col IV, PAI-1, CTGF, α-SMA, and fibronectin (Figure 3). Again, cell toxicity from the vardenafil was excluded by an XTT viability assay (Appendix A).

We further evaluated whether this occurred through canonical TGF-β1-induced SMAD signaling in AKR-2B lung fibroblasts [26]. The fibroblasts treated in vitro with increasing doses of vardenafil showed no significant changes in phospho-SMAD2 or 3 with the addition of vardenafil (Figure 4). While these data demonstrated that vardenafil can effectively reduce ECM related protein accumulation in TGFβ1-stimulated fibroblasts, they further suggest that vardenafil potentially acts downstream of SMAD2 or 3 through an SMAD-independent mechanism(s).

### 3.4. Vardenafil Treatment Suppressed the mTORC2 Signaling Pathway but Not mTORC1 Signaling

In light of the lack of effect of vardenafil on canonical SMAD signaling, we next used western blotting to assess the predominant non-SMAD pathways that might be inhabited by vardenafil in fibroblasts. We demonstrated that vardenafil pretreatment suppressed the mTORC2 signaling pathway but not the mTORC1 pathways. To address this, AKR-2B fibroblasts were cultured with various vardenafil concentrations in the absence or presence of TGF-β, and western blot analysis was performed for the various signaling proteins (Figure 5). We observed that vardenafil pretreatment of the fibroblasts showed stronger pharmacodynamic decreases in mTORC2 signaling. Specifically, pAkt (Ser473), a target of mTORC2, was decreased upon vardenafil treatment compared with mTORC1. This was in contrast to pS6K (Thr389) and p4E-BP1 (Ser65), which were targets of mTORC1 and showed no change or increase upon vardenafil pretreatment. Consistent with this finding, rapamycin, an inhibitor of mTORC1 pathways, did not alter the antifibrotic effects of vardenafil in the cultured fibroblasts (Appendix A). These data suggest that vardenafil pretreatment is sensed by the mTORC2 signaling pathway.

### 3.5. Vardenafil Inhibited TGF-β1-Mediated Fibronectin ECM Production in a Dose-Dependent Fashion and Exhibited Synergistic Inhibition of Fibronectin Production in Combination with Nintedanib in AKR-2B Fibroblasts

Using the individual dose response curves for vardenafil and nintedanib in terms of fibronectin suppression via ELISA (Appendix A), the ED_50_ for each drug was estimated using the AKR-2B fibroblast line. The ED_50_ was defined as the doses creating an effect level of 50% maximum effect of each drug observed. The experiments were repeated six times to obtain dose response curves. For AKR-2B fibroblasts, the vardenafil ED_50_ was estimated to be 12.5 µM, and the nintedanib ED_50_ was estimated to be 500 nM.

Half of the ED_50_ dose of each drug was next used in the combination experiment to test for synergy. Using this method, a dose combination of 6.25 µM vardenafil and 250 nM nintedanib was selected for experimentation. An additive drug effect would demonstrate a fibronectin value between the ED_50_ of both drugs, given the dose reduction in one drug would be supplemented by the effect of the other drug. Greater fibronectin inhibition in the dose-reduced combination group than that obtained at the respective ED_50_ doses of vardenafil and nintedanib alone would be demonstrative of a synergistic combination. Compared with the TGF-β1-stimulated control, both vardenafil and nintedanib individually resulted in significant reductions in TGF-β1-induced fibronectin expression (Figure 6A). However, in addition, the combination of vardenafil (6.25 µM) and nintedanib (0.25 µM) resulted in statistically greater suppression of fibronectin production than either dose of the individual agents alone. Given that the fibronectin suppression in the combination sample of vardenafil and nintedanib was significantly greater than the fibronectin suppression seen in the estimated ED_50_ samples of both vardenafil and nintedanib individually, the combination was judged as synergistic for the suppression of ECM fibronectin in AKR-2B fibroblasts. Cellular viability under all pretreatment conditions was confirmed by an XTT assay, with all groups demonstrating at least 100% of the viability of the control (Appendix A).

### 3.6. Vardenafil Inhibited TGF-β1-Mediated Fibronectin Production in a Dose-Dependent Fashion and Exhibited Synergistic Inhibition of In Vitro Fibronectin Production in Combination with Nintedanib in Human IPF Fibroblasts

Similar studies were also performed in human IPF fibroblasts after synergy was first demonstrated in the AKR-2B cells. Using the individual dose response curves for vardenafil and nintedanib in terms of fibronectin suppression (Appendix A), an estimated ED_50_ for each drug was identified in the human IPF fibroblast cells. The experiments were repeated nine times to obtain dose response curves. In human IPF fibroblasts, the vardenafil ED_50_ was estimated to be 25 µM and, the nintedanib ED_50_ was estimated to be 500 nM.

As was performed above, combination doses of the drug were obtained by using half of the ED_50_ doses of vardenafil and nintedanib to obtain a combination dose of 12.5 µM vardenafil and 250 nM nintedanib in the human IPF cells. The experiments were repeated seven times. As observed in the AKR-2B fibroblasts, the fibronectin production in the human IPF fibroblasts was significantly reduced individually by both vardenafil and nintedanib alone following TGF-β1-induced fibronectin (Figure 6B). Once again, in addition, the combination of vardenafil (12.5 µM) and nintedanib (0.25 µM) resulted in statistically greater suppression of fibronectin production than with either dose of the individual agents alone. Since the relative expression of fibronectin in the combination vardenafil and nintedanib sample was significantly lower than the estimated ED_50_ dose of either vardenafil or nintedanib individually, the combination of vardenafil and nintedanib was found to be synergistic for the suppression of ECM fibronectin in human disease IPF fibroblasts. Cellular viability under all pretreatment conditions was confirmed by an XTT assay, with all groups being assessed as greater than 88% of the viability of the control (Appendix A).

### 3.7. Vardenafil Reduced Collagen Type I (COL1A1) Deposition in the Lung and Increased Overall Survival in Bleomycin-Treated Mice

We next evaluated the potential effects of vardenafil in a bleomycin model of lung injury and fibrosis over 19 days, with 10 mice in each group. The overall survival curves are demonstrated in Figure 7A. Overall survival between the bleomycin/saline, bleomycin/vardenafil (20 mg/kg), and saline/saline groups was compared and found to be statistically different by a log rank (Mantel–Cox) test (*p* = 0.0155). After 19 days post bleomycin injury, collagen type I, alpha 1 (COL1A1) was measured via qPCR. Of note, the relative expression of COL1A1 was significantly higher in the bleomycin cohort, as anticipated. COL1A1 expression was significantly lower in the bleomycin mice treated with vardenafil (20 mg/kg) compared with the bleomycin/saline cohort (*p* = 0.0066) (Figure 7B). The histological examination revealed substantial restoration of the lung architecture in the presence of vardenafil pretreatment (Figure 7C). In addition, picrosirius red and Masson’s trichrome staining revealed reduced collagen and matrix expression in the animals treated with vardenafil (Figure 7C), and the hydroxyproline content (Figure 7D) was also significantly reduced in the vardenafil-treated animals. The hydroxyproline was measured as we previously reported [22]. These data support a potential beneficial role for the activity of vardenafil in lung fibrosis. To further quantity the overall fibrosis in the lung histopathology, we used a morphology score, where one indicated no fibrosis, one showed occasional small subpleural foci, two demonstrated interalveolar septal thickening and subpleural foci, and three indicated prominent and continuous interalveolar and subpleural fibrosis [22] (Figure 7E). Using this approach, we again demonstrated a significant reduction in fibrosis in mice treated with BLM and vardenafil compared with the mice receiving bleomycin alone. Overall, we demonstrated the effect of vardenafil pretreatment in the bleomycin model through the expression of collagen 1A, histology, and hydroxyproline and blinded morphology scoring. While the measured hydroxyproline levels were relatively low, perhaps being related to dilution and extraction, all measures of the vardenafil pretreatment effect consistently showed a benefit with vardenafil.

Finally, additional studies on the bleomycin-treated mice were undertaken to determine whether synergy could also be observed in the bleomycin animal models. Unfortunately, we did not observe significant differences in the effects of the combination of vardenafil with or without nintedanib in this animal model on the suppression of fibrosis as measured by extracellular matrix generation (Appendix A). After 19 days post bleomycin injury, there was significant generation of extracellular fibronectin in the bleomycin-treated mice, which was significantly suppressed by either vardenafil (10 mg/kg) or nintedanib (30 mg/kg). However, the combination of the two agents did not demonstrate any additional significant benefit (*p* = 0.26). Thus, while we observed in vitro synergy between vardenafil and nintedanib, in vivo synergy was not demonstrated.

## 4. Discussion

In this study, we observed that the PDE5-I vardenafil successfully reduced matrix mRNA and ECM protein expression, markers of EMT, and anchorage-independent growth in cultured lung fibroblasts. Vardenafil independently reduced matrix fibronectin production in AKR-2B and human IPF fibroblasts and exhibited synergistic suppression of fibronectin generation when combined with nintedanib. In addition, we observed improved survival and the matrix protein profiles in the bleomycin-challenged mice treated with vardenafil. The improved survival may be related in part to reduced fibrosis, as well as the vascular and other potential beneficial effects of vardenafil. Together, our data suggest that vardenafil may exhibit potential activity in IPF, with potential benefits beyond its established effects on the pulmonary vasculature to ameliorate pulmonary fibrosis. However, these observations are certainly early, and significant additional preclinical work will be required before these agents can be applied to humans.

The anti-fibrotic role of PDE-I has been demonstrated in other fibrotic models. In a rodent model of anti-Thy1 antibody-induced mesangial proliferative glomerular-nephritis, the PDE-I vardenafil decreased the TGF-β1 activating protein TSP-1 and the TGF-β1 signaling protein *p*-SMAD-2/3, indicating direct antifibrotic activities for this PDE-I [23,24]. Downregulation of cAMP due to increased PDE4 expression in renal tubular epithelial cells was found in the fibrotic renal tissue and was ameliorated by replenishing cAMP via inhibition of PDE4 [27]. PDE1A has also been shown to play an important role in cardiac hypertrophy and fibrosis in cardiac fibroblasts [28]. In addition, in the mouse bleomycin-induced pulmonary fibrosis model, data suggest that the PDE5-I sildenafil may also ameliorate pulmonary fibrosis following drug-induced lung injury [29]. Anti-fibrotic effects have been demonstrated with PDE4 inhibition in bleomycin mice as well [30]. Another PDE5-I, tadalafil, has been found to protect against silica-induced pulmonary fibrosis in a rodent model of silicosis, with evidence of decreased TGF-β1, collagen, and other ECM components [31]. Our data strongly support a potential role for vardenafil in lung fibrosis, and it has the added potential benefit of synergy, based on our in vitro analysis. The bleomycin model presents numerous challenges, among them being testing combinations of agents to yield significant and reliable results. Additional animal studies were undertaken in an attempt to evaluate whether synergy could also be observed in vivo using the bleomycin animal model. Unfortunately, we did not observe significant differences in the effects of the combination of vardenafil with or without nintedanib in this animal model. This could be the result of accessible dosing or the limitations of the model, or it could represent a lack of in vivo synergy in this model. Additional studies of other animal models of lung fibrosis will be required to establish the preclinical benefit prior to investigations in humans.

These findings in other models demonstrate the activity of vardenafil as a potential anti-fibrotic agent. The side effects of nintedanib include nausea, vomiting, arterial thrombosis, and liver injury, which may be poorly tolerated in the IPF population. Opportunities for dose reduction in the presence of other novel anti-fibrotic agents may allow for attenuation of these side effects, a reduction in toxicity, and the optimization of anti-fibrotic therapeutic potential. Recently, Kolb et al. reported a randomized clinical trial of nintedanib plus sildenafil, another PDEI, in patients with idiopathic pulmonary fibrosis and a DLCO of 35% or less. Notably, they did not observe a difference in the St. George’s Respiratory Questionnaire scores with the addition of sildenafil to nintedanib compared with nintedanib alone over 12 weeks [32]. Of interest, in vitro testing of sildenafil in our assay systems did not reveal the same anti-fibrotic activity that we observed for vardenafil (Appendix A). Nonetheless, it will be important to eventually test such potential treatment combinations in patients over adequate time frames to determine whether these agents have activity in altering disease activity or progression.

There are several limitations to our study. First, our study was primarily focused on fibroblast activity, since these are the predominant cells found in the fibroblastic foci of IPF. However other cells certainly participate in the fibroproliferative repair process in this disease, including epithelial cells and macrophages, as well as the process of epithelial mesenchymal transformation [3,4,6,7]. In addition, IPF is a disease strongly associated with aging, and cellular senescence also participates in the overall disease pathogenesis [11]. It remains uncertain whether vardenafil affects the other cells or pathogenetic processes. Such studies were beyond the scope of the current investigation, and further studies will be required to delineate these other cellular processes.

In addition, our animal studies were undertaken for a proof of concept, with the vardenafil and test agents being administered as pretreatments shortly after the bleomycin. As such, the studies demonstrated activity in prevention and therapy. However, prior to use in humans, additional preclinical studies will be required, such as with the initial treatment of bleomycin and then later treatment with the test agents after the establishment of fibrosis. Nevertheless, our studies do provide support for the potential activity of vardenafil in lung fibrosis.

In conclusion, our study demonstrates the activity of vardenafil as a potential antifibrotic agent, as well as in vitro synergistic combination with the FDA-approved anti-fibrotic nintedanib. However, in vivo synergy of the agents in the mouse models of lung fibrosis was not demonstrated. Additional studies will be required to investigate the utility of PDE5-I inhibitors such as vardenafil in human subjects and elucidate additional mechanisms, by which their effects can be attained.

## Figures and Tables

**Figure 1 cells-10-03502-f001:**
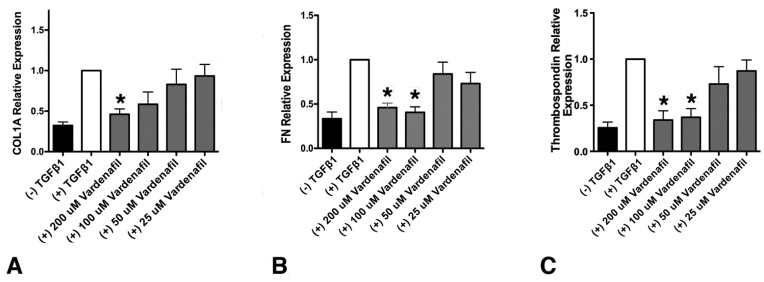
Vardenafil inhibited TGF-β1-mediated, matrix-associated mRNA levels in lung fibroblasts. The effect of the respective mRNA transcripts in human IMR-90 fibroblasts stimulated with TGF-β1 after 24 h in the presence of the indicated vardenafil concentrations are shown. (**A**) Collagen type I, alpha 1 (COL1A1), (**B**) fibronectin, and (**C**) thrombospondin (TSP-1) relative mRNA expression in IMR-90 lung fibroblasts. Data are from six experimental runs and are expressed as means ± SEM. * *p* < 0.05, comparing the tested condition to the control maximal stimulation in the presence TGF-β1 but in the absence of vardenafil.

**Figure 2 cells-10-03502-f002:**
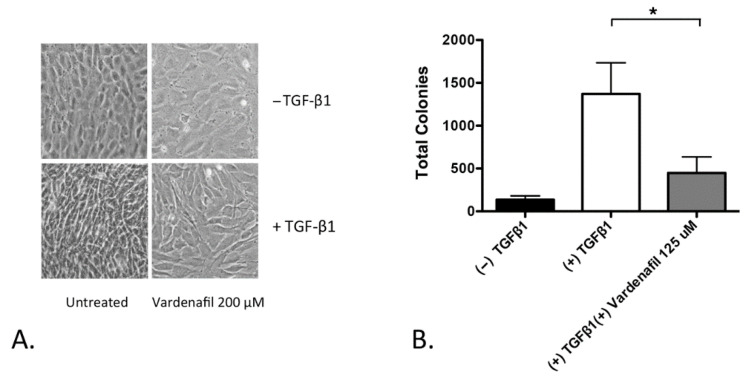
Vardenafil inhibited morphological transformation and anchorage-independent growth (AIG) of fibroblasts. (**A**) Photographs of representative fields are shown with indicated pretreatments in AKR-2B fibroblasts, demonstrating that TGF-β1 induced morphological changes, which were suppressed in the presence of vardenafil. (**B**) In addition, AKR-2B soft agar anchorage independent growth experiments demonstrate that vardenafil significantly reduced the numbers of anchorage-independent colonies formed during culture. The data represent the mean ± SEM. * *p* < 0.05 on the comparison shown.

**Figure 3 cells-10-03502-f003:**
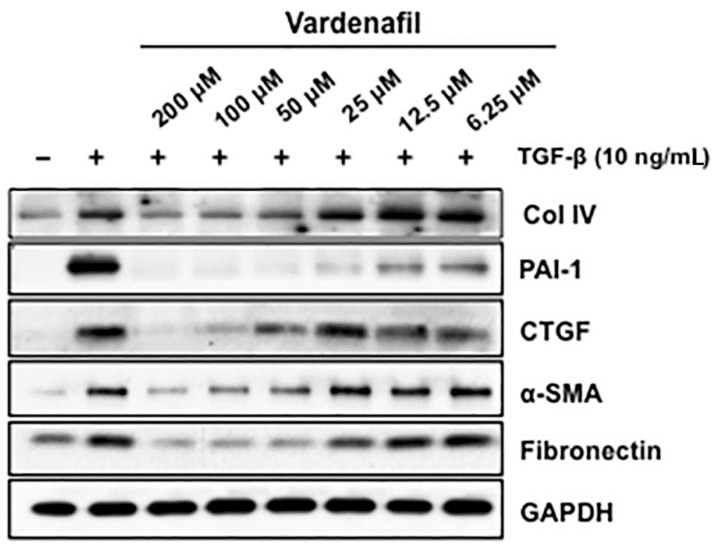
Vardenafil inhibited matrix-associated protein generation by fibroblasts. Quiescent AKR-2B fibroblasts in 0.1% FBS/DMEM were pretreated for 60 min with the indicated vardenafil concentration (PDE5 inhibitors). Vehicle (−) or TGF-β (+) was directly added to a final concentration of 10 ng/mL, and following 24 h of incubation, lysates were prepared and western blotted for Collagen I, PAI-1, CTGF, α-SMA, and fibronectin. GAPDH was used as a loading control. Data are representative of three separate experiments.

**Figure 4 cells-10-03502-f004:**
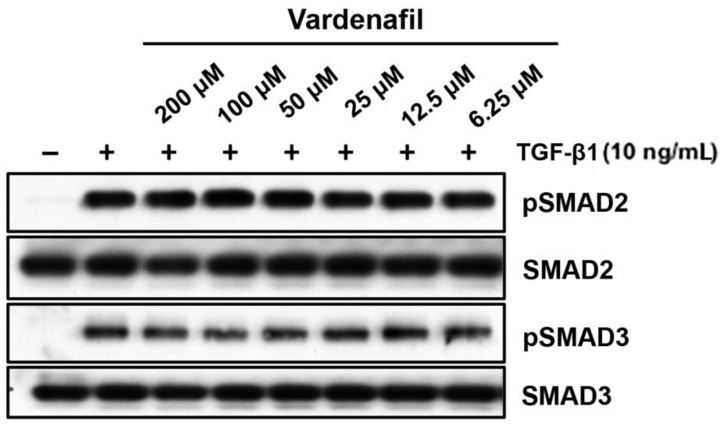
Vardenafil had little or no effect on TGF-β1-driven phosphorylation of SMAD2 and SMAD3. To determine whether vardenafil suppressed classical TGF-β1-driven SMAD signaling, western blottings of AKR-2B samples were harvested after 6 h, and pSMAD2, SMAD2, pSMAD3, and SMAD3 were determined in the presence and absence of TGF-β1 pretreatment in the presence of different vardenafil concentrations. Shown are the data representative of two separate experiments.

**Figure 5 cells-10-03502-f005:**
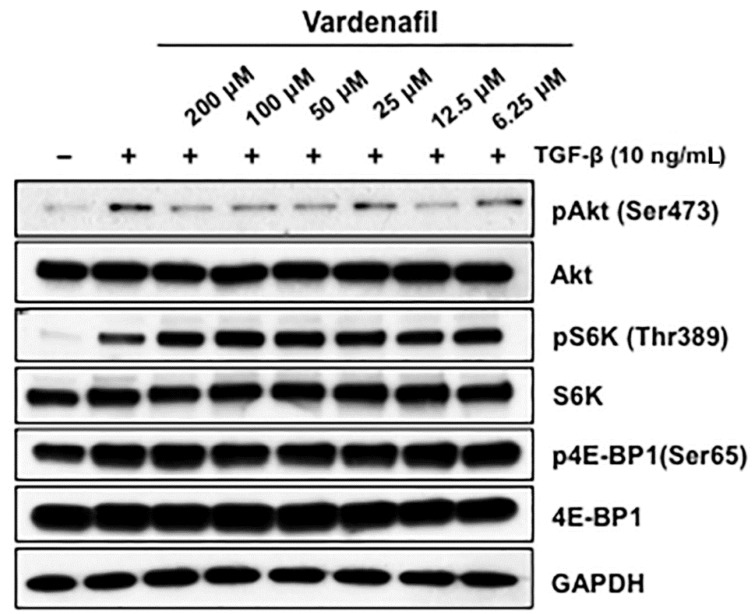
Vardenafil pretreatment suppressed the mTORC2 signaling pathway but not mTORC1. Fibroblasts (AKR-2B) in 0.1% FBS/DMEM were cultured with various vardenafil concentrations and were treated in the absence (−) or presence (+) of TGF-β (10 ng/mL). Following 6 h of stimulation, western blot analysis was performed for the indicated signaling proteins. Data are representative of three separate experiments. Vardenafil pretreatment of the fibroblasts showed stronger pharmacodynamic decreases in mTORC2 signaling, since pAkt (Ser473), which was a target of mTORC2, was decreased upon vardenafil treatment compared with mTORC1. In contrast, pS6K (Thr389) and p4E-BP1(Ser65), which were targets of mTORC1, showed no change or increase upon vardenafil treatment, suggesting that vardenafil pretreatment is sensed by the mTORC2 signaling pathway.

**Figure 6 cells-10-03502-f006:**
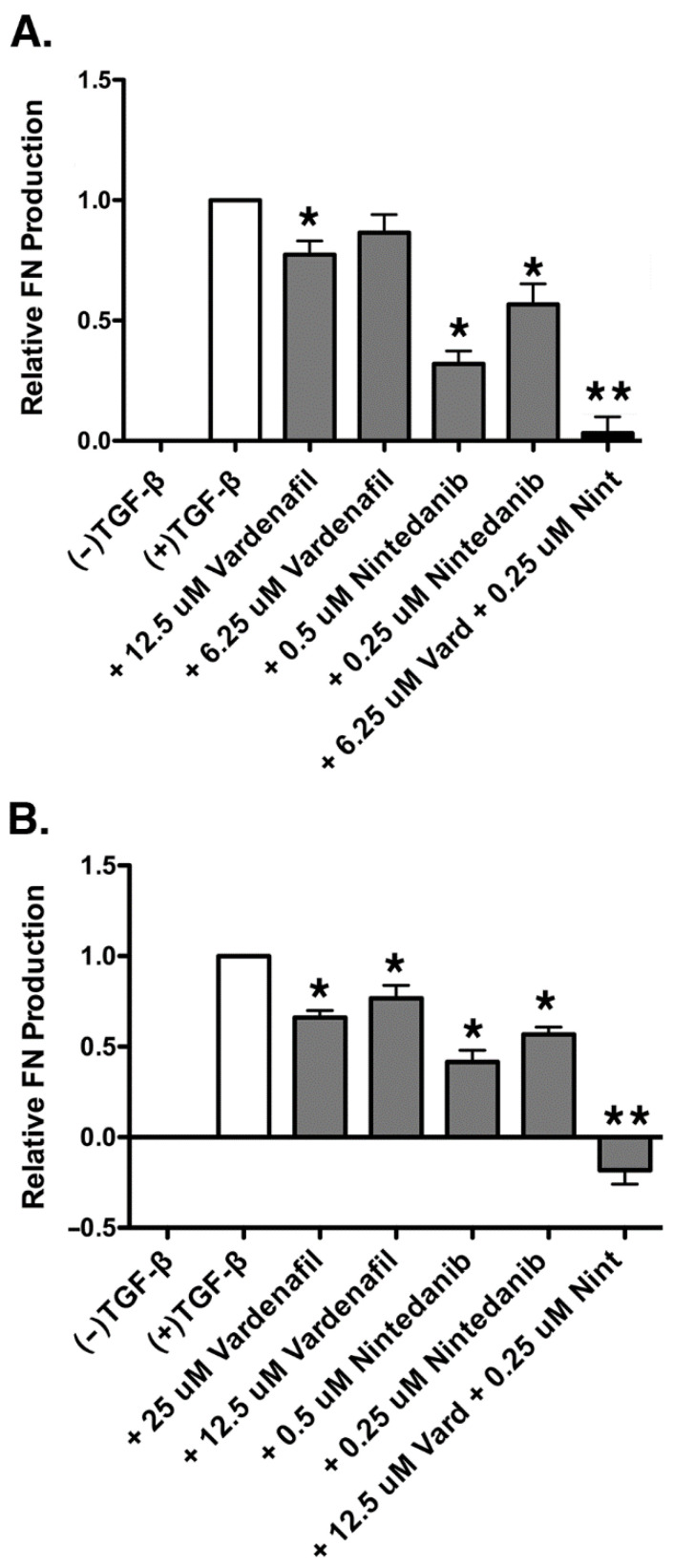
Vardenafil inhibited TGF-β1-mediated fibronectin production by AKR-2B and human IPF fibroblasts and exhibited synergistic inhibition of fibronectin production in combination with nintedanib. (**A**) Fibronectin production was assessed via ELISA in AKR-2B fibroblasts stimulated with TGF-β1 and treated with a synergistic combination of vardenafil and nintedanib. Shown are the total of six experiments with data expressed as the mean ± SEM. * *p* < 0.05 compared with +TGF-β1 control. ** *p* < 0.05 compared with all other conditions. (**B**) Fibronectin production was assessed via ELISA in IPF fibroblasts stimulated with TGF-β1 and treated with a synergistic combination of vardenafil and nintedanib. Shown are data from seven experiments with results expressed as the mean ± SEM. * *p* < 0.05 compared with +TGF-β1 control. ** *p* < 0.05 compared with all other conditions.

**Figure 7 cells-10-03502-f007:**
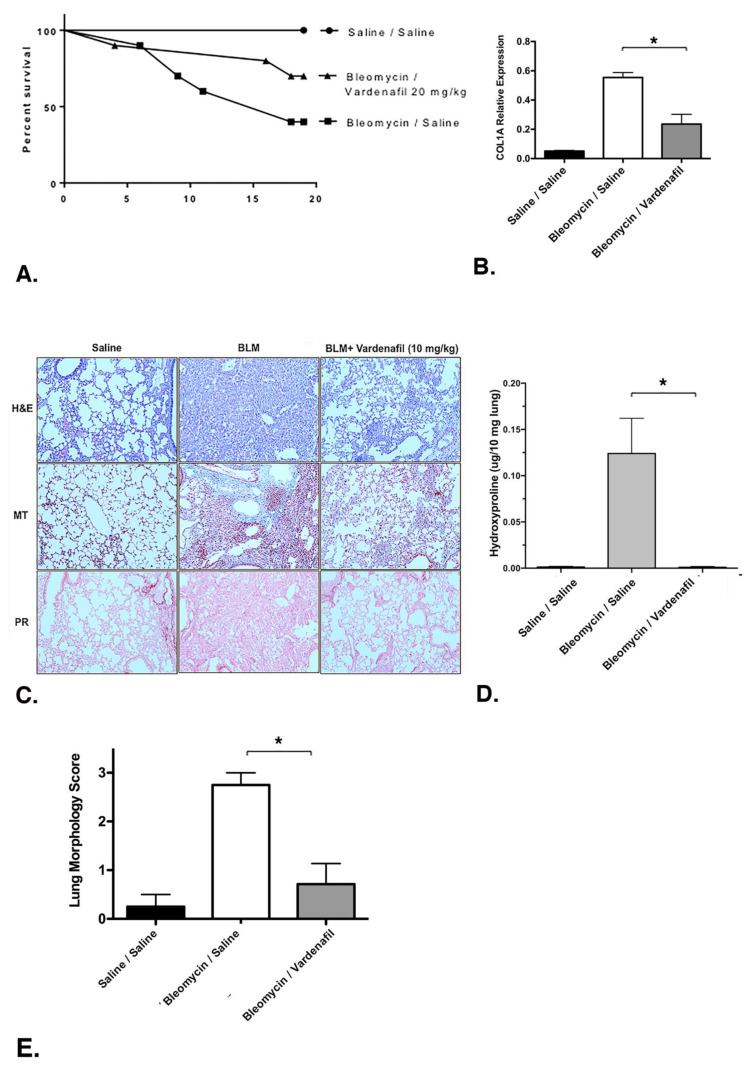
Vardenafil improved survival and reduced collagen production in a bleomycin model of lung injury and fibrosis. The mice received intratracheal bleomycin and were treated with vardenafil over 19 days, as detailed under Materials and Methods. The size of each group at the time of injury was 10. (**A**) Survival curves for the saline/saline, bleomycin/saline, and bleomycin/vardenafil (20 mg/kg) mouse groups are demonstrated. Vardenafil pretreatment of bleomycin injured mice resulted in significantly better survival compared with the bleomycin-treated mice not receiving vardenafil (*p* = 0.0155 by a log rank Mantel–Cox test). (**B**) Vardenafil pretreatment was also associated with reduced collagen 1A expression. The relative expression of COL1A mRNA was assessed via qPCR. * *p* < 0.05 for comparing lung COL1A mRNA levels in the bleomycin-treated mice with or without vardenafil treatment. (**C**) Histological examination of the lungs from these animals revealed substantial restoration of the lung architecture in the bleomycin-treated animals that also received vardenafil pretreatment (H + E = hematoxylin and eosin). In addition, the picrosirius red (PR) and Masson’s trichrome (MT) staining revealed reduced collagen and matrix expression in the bleomycin-treated mice who also received vardenafil. (**D**) The lung hydroxyproline content was also significantly reduced in the vardenafil-treated animals. * *p* < 0.05 when comparing lung hydroxyproline levels in the bleomycin-treated mice with or without vardenafil treatment. (**E**) The lung histopathology sections were also blindly scored for the degree of fibrosis using a method we previously reported [22]. Using this approach, we again demonstrated a significant reduction in fibrosis in mice pretreated with BLM and vardenafil compared with mice receiving bleomycin alone. **p* < 0.05 when comparing lung morphology scores in the bleomycin-treated mice with or without vardenafil treatment.

## Data Availability

Data is contained within the article or Appendix A.

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
