# Peer review of "Vardenafil Activity in Lung Fibrosis and In Vitro Synergy with Nintedanib"

_cells, 2021, doi:10.3390/cells10123502_

Round 1

Reviewer 1 Report

The authors investigated the synergic effect of verdanefil and nintedanib on fibroblast TGF-β1 driven extracellular matrix through fibronectin ELISA and in a bleomycin mouse model. The results are very interesting and worthworthly to deep study.

  1. Abstract: please cyte the acronym in full the first time you use them (such as PDE5, TGF, ECM, ELISA)
  2. Materials, methods and results were appropriately described
  3. Please report the limitations of the study in discussion section

Author Response

Response to General Comments:  We appreciate the very supportive comments of the reviewer and have made the specific comments that follow.

Comment 1. Abstract: please cite the acronym in full the first time you use them (such as PDE5, TGF, ECM, ELISA)

Response to Comment 1: We appreciate the suggestion of the reviewer and have defined the acronyms at first usage throughout the revised manuscript.

Comment 2. Materials, methods and results were appropriately described

Response to Comment 2:  We appreciate the supportive comments. No modification required.

Comment 3. Please report the limitations of the study in discussion section

Response to Comment 3:  We agree with the reviewer’s suggestions and have added a paragraph covering the limitations of the study in the discussion on lines 400-407 of the revised manuscript.

Reviewer 2 Report

This paper shows that vardenafil has an inhibitory effect on fibrotic pathways mainly in vitro. However, several flaws impede its publication in the present form.

1 IPF physiopathology was no mentioned PubmedID: 21719092. Especially by the fact that epithelial cells were not considered and have a protagonist role.

2 Another omitted aspect is that IPF is an age-related disease. Thus, the lines used are not apt as representative controls.

3 In a similar way, the pathways evaluated are activated by the effect of aging that it has been previously reported (mTOR) PMID: 27566137.  

4 Title don’t represent the results that could be vardenafil prevents myofibroblast differentiation by TGF-beta.

5 Finally, usually hydroxyproline quantification in the bleomycin model should be on the left whole lung.

Author Response

Comment 1. IPF physiopathology was no mentioned PubmedID: 21719092. Especially by the fact that epithelial cells were not considered and have a protagonist role.

Response to Comment 1:  We agree with the reviewer’s suggestion and have included a paragraph discussing the limitations of the current studies (lines 400-408) of the revised manuscript.  In that paragraph we discuss the potential roles of epithelial cells and macrophages in the overall disease pathogenesis.

Comment 2. Another omitted aspect is that IPF is an age-related disease. Thus, the lines used are not apt as representative controls. 

Response to Comment 2:  We agree with the suggestion and in the additional paragraph (lines 400-408) of the revised manuscript, have discussed the role of aging in the pathogenesis of IPF.

Comment 3. In a similar way, the pathways evaluated are activated by the effect of aging that it has been previously reported (mTOR) PMID: 27566137.  

Response to Comment 3 : Please see the response to comment 2 above.  We have now included reference 11 in the revised discussion.  That reference does nicely review the effects of aging in IPF pathogenesis.

Comment 4. Title don’t represent the results that could be that vardenafil prevents myofibroblast differentiation by TGF-beta.

Response to Comment 4:  We appreciate the reviewer’s comments.  We believe that vardenafil is impacting the fibroproliferative process through a variety of activities. These include direct effect on matrix production, fibroblast morphologic changes (myofibroblast differentiation), anchorage independent growth, and direct effects on cell signaling. This is reflected in figures 2, 3, 4, and 5 of the revised manuscript. Because of these multiple activities in fibrosis, we prefer to leave the manuscript title general as we originally proposed. However, if the editor wishes us to change the title, we will be happy to do so.

Comment 5. Finally, usually hydroxyproline quantification in the bleomycin model should be on the left whole lung.

Response to Comment 5:  We performed the hydroxyproline as described in our previously published investigations [22].  This is noted on line 326 of the revised manuscript.

Reviewer 3 Report

In this study, Bourne and co-authors examined the PDE5-I vardenafil activity alone and in synergy with nintedanib in vitro, in cultured lung fibroblasts as well as in vivo in bleomycin-induced model of lung fibrosis. The results in the in vivo model did not confirm any synergistic effect, however. It is an interesting study, coherent, well written, in which authors examined the anti-fibrotic action of vardenafil in diverse levels combining a variety of technics. I am concerned about some minor issues.

Minor:

  1. In one group mice received intratracheal bleomycin and treated with vardenafil over 19 days. I would like to ask why you chose to treat with vardenafil from the beginning of the inflammatory phase within the first 2 weeks after the injury and not later during the fibrotic phase. There is some evidence suggesting that targeting mechanisms occurring during the inflammatory phase are not likely to interpret into a clinical benefit and that in order to determine new drugs efficacy their testing should be performed later during the fibrotic phase.

Author Response

General Comments: In this study, Bourne and co-authors examined the PDE5-I vardenafil activity alone and in synergy with nintedanib in vitro, in cultured lung fibroblasts as well as in vivo in bleomycin-induced model of lung fibrosis. The results in the in vivo model did not confirm any synergistic effect, however. It is an interesting study, coherent, well written, in which authors examined the anti-fibrotic action of vardenafil in diverse levels combining a variety of technics. I am concerned about some minor issues.

Response to the General Comments:  We greatly appreciate the supportive comments of the reviewer and have addressed the minor comments which follow.  We have also discussed our observations that we did observe synergy with vardenafil and nintedanib in the cell systems in vitro but were not able to show synergy in our animal model.  This is discussed on lines 384-389 of the revised manuscript.

Minor: 1. In one group mice received intratracheal bleomycin and treated with vardenafil over 19 days. I would like to ask why you chose to treat with vardenafil from the beginning of the inflammatory phase within the first 2 weeks after the injury and not later during the fibrotic phase. There is some evidence suggesting that targeting mechanisms occurring during the inflammatory phase are not likely to interpret into a clinical benefit and that in order to determine new drugs efficacy their testing should be performed later during the fibrotic phase

Response to Minor Comment 1:  We fully appreciate the comment of the reviewer and agree.  We have inserted a discussion of this issue in the limitations portion of the discussion (line 414-419).  “Our animal studies were undertaken for proof of concept, with the vardenafil and test agents being administered shortly after the bleomycin. As such, the studies demonstrate activity in prevention and therapy. However, prior to use in humans, additional preclinical studies will be required, such as with initial treatment of bleomycin and then later treatment with the test agents after the establishment of fibrosis. Nevertheless, our studies do provide support for the potential activity of vardenafil in lung fibrosis.”

Round 2

Reviewer 1 Report

The authors appropriately responded to all of comments.

Author Response

We appreciate the effort by the reviewer. 

Reviewer 2 Report

In this new version of the article, the lack of controls persists and it is necessary to redo experiments to be able to consider the conclusions.

The introduction has to include a paragraph on the pathogenesis of idiomatic pulmonary fibrosis not only in the discussion. 

In vitro experiments still lack adequate age controls and should at least be mentioned in the text.

In addition, figures 4 and 5 have to show the baseline comparison of IPF cells vs control in the same blot. Since the mentioned pathways are not active in the IPF lines that were studied (more IPF lines could help). 

The hydroxyproline levels in the model show a very low degree of hydroxyproline so it doesn't seem to have worked. In addition, the measurement of hydroxyproline is not adequate, they are based on a 1999 protocol, and considering these values suggests that vardenafil intervenes with the inflammatory phase and not with the fibrosis phase. Therefore the results cannot be interpreted as the authors suggest.

Vardenafil does not have enough evidence to start a clinical study as suggested in this version and this should be removed in the text.
